# Using Transcriptomic Analysis to Assess Double-Strand Break Repair Activity: Towards Precise in Vivo Genome Editing

**DOI:** 10.3390/ijms21041380

**Published:** 2020-02-18

**Authors:** Giovanni Pasquini, Virginia Cora, Anka Swiersy, Kevin Achberger, Lena Antkowiak, Brigitte Müller, Tobias Wimmer, Sabine Anne-Kristin Fraschka, Nicolas Casadei, Marius Ueffing, Stefan Liebau, Knut Stieger, Volker Busskamp

**Affiliations:** 1Center for Regenerative Therapies (CRTD), Technical University Dresden, 01307 Dresden, Germany; 2Institute of Neuroanatomy & Developmental Biology (INDB), Eberhard Karls University Tübingen, 72074 Tübingen, Germany; 3Department of Ophthalmology, Justus-Liebig-University, 35392 Giessen, Germany; 4Institute of Medical Genetics and Applied Genomics, University of Tübingen, 72076 Tübingen, Germany; 5DFG NGS Competence Center Tübingen, 72076 Tübingen, Germany; 6Department of Ophthalmology, Institute for Ophthalmic Research, University of Tübingen, 72076 Tübingen, Germany; 7Universitäts-Augenklinik Bonn, University of Bonn, Dept. of Ophthalmology, 53127 Bonn, Germany

**Keywords:** inherited retinal dystrophies, double-strand breaks, DSB repair pathways, single-cell RNA-seq, hiPSC-derived retinal organoids, photoreceptors

## Abstract

Mutations in more than 200 retina-specific genes have been associated with inherited retinal diseases. Genome editing represents a promising emerging field in the treatment of monogenic disorders, as it aims to correct disease-causing mutations within the genome. Genome editing relies on highly specific endonucleases and the capacity of the cells to repair double-strand breaks (DSBs). As DSB pathways are cell-cycle dependent, their activity in postmitotic retinal neurons, with a focus on photoreceptors, needs to be assessed in order to develop therapeutic in vivo genome editing. Three DSB-repair pathways are found in mammalian cells: Non-homologous end joining (NHEJ); microhomology-mediated end joining (MMEJ); and homology-directed repair (HDR). While NHEJ can be used to knock out mutant alleles in dominant disorders, HDR and MMEJ are better suited for precise genome editing, or for replacing entire mutation hotspots in genomic regions. Here, we analyzed transcriptomic in vivo and in vitro data and revealed that HDR is indeed downregulated in postmitotic neurons, whereas MMEJ and NHEJ are active. Using single-cell RNA sequencing analysis, we characterized the dynamics of DSB repair pathways in the transition from dividing cells to postmitotic retinal cells. Time-course bulk RNA-seq data confirmed DSB repair gene expression in both in vivo and in vitro samples. Transcriptomic DSB repair pathway profiles are very similar in adult human, macaque, and mouse retinas, but not in ground squirrel retinas. Moreover, human-induced pluripotent stem-cell-derived neurons and retinal organoids can serve as well suited in vitro testbeds for developing genomic engineering approaches in photoreceptors. Our study provides additional support for designing precise in vivo genome-editing approaches via MMEJ, which is active in mature photoreceptors.

## 1. Introduction

Inherited retinal diseases (IRDs) are a group of disorders with a prevalence of 1 in 3-4000 people [1]. Due to the complexity of the visual system, several hundred proteins are uniquely expressed in photoreceptors and retinal pigment epithelium (RPE) cells, and mutations in over 200 genes have been associated with IRDs [2]. Over the past decades, advancements in gene addition therapy have demonstrated significant success in treating specific forms of IRD, such as RPE65 deficiency [3]. Specific advantages of the eye, such as being readily accessible, highly compartmentalized and immune-privileged, have positioned this organ at the forefront of gene therapy development [4].

Yet, many IRDs, such as Stargardt disease associated with mutations in the ABCA4 gene, cannot be treated with gene supplementation therapy: The underlying genes do not meet the cargo size requirements of state-of-the-art and approved viral vector systems (Figure 1a); and dual adeno-associated virus (AAV) vector systems, on the other hand, are not efficient enough [5,6]. Moreover, gene supplementation cannot be used when the retinal degeneration is caused by mutations in dominant genes [7,8]. To this end, in vivo genome editing represents a promising emerging treatment of such monogenic disorders, as it aims to correct the disease-causing mutation within the genome, resulting in the restoration of endogenous protein production [9]. To target a large number of disease-causing mutations within the same gene, replacing large DNA fragments comprising several exons might be advantageous over targeting single mutations (Figure 1c) [10]. 

Therapeutic genome editing applications are based on the specific induction of DNA double-strand breaks (DSB) and the cell’s capacity to repair them to maintain genomic stability [11]. Three pathways can repair a DSB: Non-homologous end-joining (NHEJ) or, in the presence of a DNA template, homology-directed repair (HDR) and microhomology-mediated end-joining (MMEJ) [12]. Each of these pathways requires many different DNA repair factor sets [13]. NHEJ is the predominant DSB repair pathway at all cell-cycle steps. Its flexibility enables a wide range of DNA end configurations to be repaired, and it often results in mutations, insertions, or deletions (i.e., indels) at repaired DNA junctions [14]. HDR is the most precise pathway, as it results in high-fidelity DSB repair. It is mostly regulated by cycle-dependent cytokines and it uses the sister chromatid as a template for repair, therefore it naturally occurs in the S and G2 phases of the cell cycle [15], [16]. MMEJ, also called alternative end-joining (a-EJ), relies on a different set of proteins than HDR and NHEJ. It requires microhomology regions (5-25bp) to donor DNA strings at the overhangs of DNA ends, and it has been reported to be more active when either HDR or NHEJ are inactive. After a DSB is introduced, the protection of the broken ends can initiate NHEJ or, if an initial 3’ or 5’ strand resection occurs, the HDR repair machinery can be recruited (Figure 1b) [14]. For NHEJ, the Ku complex protecting the DNA ends (Ku70-Ku80) recruits specific ligases and nucleases that tether the DNA ends. If DNA-strand resection is initiated, the MRN complex is recruited to the DSB: MMEJ then occurs in the presence of donor DNA microhomology and binding of PARP1. HDR or secondary-strand annealing (SSA) can repair the DSB, when the resection is extended by other nucleases, and other determinant proteins, such as RAD51 and BRCA1 for HDR, bind to the overhangs.

Most of the existing knowledge about exploiting DSB repair has been gathered from cell-culture studies using artificial cell lines such as HEK293 [17,18,19]. The activity state of the DSB repair machinery in highly specialized photoreceptors (PR) is currently largely unknown. There have been some studies demonstrating that it is possible to knock down protein expression via NHEJ in adult photoreceptors (PR) cells [20]. In a different study, up to 20% of NHEJ events and 2% of HDR events were observed in a mouse model containing a homing endonuclease restriction site on the X chromosome, when PR cells were targeted using an AAV vector-based approach [9]. In addition, AAV vector-based transfer of large DNA sequences can result in integration at the target locus, even in the absence of homologous regions: This method is called homology-independent targeted integration (HITI). This strategy has been shown to function at the MERTK locus in rats in vivo [21]. The aim of this study is to test whether gene expression data can be used to characterize DSB repair activity in different cell types. We employed transcriptomic data to harness the complexity of DSB repair pathways. DSB scores based on gene expression of key genes, can outline pathway activities in the transition from cycling progenitor cells to post-mitotic retinal cells. By exploiting published transcriptomic datasets, we observed conserved DSB pathway expression profiles in adult retinas of human (*Homo sapiens*), macaque (*Macaca fascicularis*) and mouse (*Mus musculus*). Then, we assessed the activity of DSB repair pathways in human photoreceptors and in relevant preclinical model organisms, a crucial step towards developing and optimizing therapeutic genome editing interventions in postmitotic photoreceptors. Rods and cones showed similar DSB pathways profiles in both human and mouse retina. We revealed similarities between in vivo and in vitro photoreceptors, developing hiPSC-derived retinal organoids and performing scRNA-seq at several time points. Using scRNA-seq, we could evaluate the cell-type composition and reconstruct lineages of differentiation of the developing retinal organoids, allowing us to evaluate DSB repair activity at different stages and across cell-types. Furthermore, we described hiPSC-derived neurons as a good in vitro model of human retinal DSB repair activity based on the DSB repair-gene expression profile.

## 2. Results

### 2.1. Homology Repair Correlates with Cell-Cycle Activity

First, we evaluated if DSB pathway activity can be extracted from gene expression data by calculating and comparing DSB scores in a cell-cycle-dependent manner. We analyzed the retinal cell type composition of mouse embryos at day 15 (E15) using a published dataset [22]. As previously shown [22], cells forming clusters were identified by known markers of progenitor cells, retinal ganglion cells, amacrine cells and PR cells (Figure 2a and Figure A1a). This dataset represents a single snapshot of the early developing retina. Still, transition stages from cycling progenitors to differentiated cell types can be captured by a uniform manifold approximation and projection (UMAP) (Figure 2a). Progenitor cells showed an organization depending on the cell-cycle stage as revealed by scoring the murine cell-cycle-associated genes (Figure A1b) [23].

To study the DSB repair pathway dynamics during the PR differentiation, a subset of cells representing the progenitors-to-PR branch (purple cluster in Figure 2a) was further characterized. This sample was reanalyzed, giving the same organization as before on a two-dimensional UMAP analysis, but it resulted in a slightly different clustering (Figure 2b). We assessed the direction of the transition progenitor-to-PR clusters using the RNA-velocity method [24]. Using the spliced and unspliced read counts, the RNA-velocity analysis computed a velocity vector for each cell, corresponding to its most likely next position on the two-dimensional UMAP embedding (Figure 2b). The clusters were similarly ordered by measuring transitions between cells through the so-called diffusion pseudotime [25]. Cells could then be sorted according to their pseudo-temporal order. The pseudotime dimension was used to evaluate dynamics in the PR trajectory (Figure A1d). A score for cells types in each of the three DSB repair pathways (NHEJ, MMEJ, and HDR) was computed using a curated gene list (see Methods 4.1). Cells of the progenitor clusters showed higher scores than differentiated cells for all the DSB repair pathways (Figure 2c). As previously reported, HDR gene expression decreased rapidly, as soon as the cells exited the cell-cycle (brown cluster in Figure 2c). Progenitors showed HDR and MMEJ scores higher than NHEJ (*t*-test, *p*-value = 7.3×10^−17^), reflecting an up-regulation of these pathways at early stages (Figure A1c). The same was observed at the neuroblast stage. The NHEJ score became significantly higher than HDR pathways at the PR stage (Figure A1c; *t*-test, *p*-value = 0.0002). 

Similarly, we conducted a time-series analysis on published bulk RNA samples of adult mouse retinas [26], CRX+ cells from in vitro retinal organoids [27], and human induced pluripotent stem cell (hiPSC)-derived neurons (inducible neurogenin cells, iNGN) [28], [29]. The mouse dataset included all retinal cell types. Organoids were dissociated and sorted for CRX+ cells for sequencing in order to capture RNA from the photoreceptor lineage. The iNGN dataset consisted of iPSC cells (uninduced at day 0) and developing neurons over time. Neuronal differentiation was induced by overexpression of the transcription factors neurogenin-1 and neurogenin-2. 

The top 20 genes associated with the G2/M phase described by Cyclebase 3.0 were analyzed [30]. As expected, all the datasets consistently showed that gene expression in the HDR and MMEJ pathways was higher when the cells were cycling; in contrast, NHEJ gene expression did not decrease suddenly when the cells stopped cycling (Figure 2d). Altogether, these results showed that the transcriptomic profile of the DSB pathways can serve as a proxy of their activity and that the human in vitro model mimicked well the retinal in vivo pathway activity.

### 2.2. Photoreceptor DSB Pathway Activity

We performed scRNA-seq (10× Genomics) to profile the cell composition across development of hiPSC-derived retinal organoids from early stages to almost one year (64, 106, 201, and 330 days), generated according to a previously described protocol [31]. Analysis of 26,700 individual cells revealed major retinal cell-types gradually appearing in time (Figure 3a). Clusters were annotated by scoring cells by cell-cycle, and by computing the overlap between t-test calculated markers and canonical retinal markers (Figure A2a–c; retinal marker genes in Table A1). Clusters plotted in 2D using t-stochastic neighbor embedding (t-sne) of each batch showed that the number of progenitor cells and cycling cells decreased over the first two time points (Figure 3a). Accordingly, increasing numbers of differentiated PR cells, Müller glia, and bipolar cells were detected at later stages. In line with previous data [31], we observed retinal ganglion cells and amacrine cells gradually disappearing (Figure 3a). One group of cells starting to appear at day 201 had high levels of NFI transcription factor expression (*NFIA*, *NFIB*, and *NFIX*): These factors have recently been described as being expressed in retinal late progenitors controlling bipolar and Müller glia differentiation [32]. For further analysis, all time points were aggregated into a single dataset. We established a two-dimensional UMAP representation of developing retinal organoids (Figure 3b,c). This representation contains a central group of progenitor cells giving rise to differentiated cell types following the first UMAP dimension. Of note, the organoids already contained cone precursors at day 64, whereas rods only started to develop consistently later, becoming the most abundant cell type at day 201. The presence of the rod and cone populations at day 330 was also confirmed by immunostaining for *ARR3* (cones) and *GNAT1* (rods) (Figure 3d). 

The DSB repair pathway scores were computed for each cell as described above (Results 2.1). Pseudotime was computed on a subset of the clusters, taking into account only progenitor cells and the neuronal cell-fate trajectory (Figure A2d). Although both HDR and MMEJ scores decreased throughout the maturation, MMEJ stayed at a consistently higher level than HDR up to post-mitotic stage (Figure 3e). On the other hand, the NHEJ score decreased only slightly from progenitors to differentiated cell types and stayed significantly higher than HR pathways in both cones and rods of hiPSC-derived organoids (Figure 3f). We observed a similar DSB profile, repeating the analysis in published scRNA-seq datasets from both human and murine adult retinas [23,33,34] (Figure A3). Together, these results highlight that there are no significant differences between rods and cones, neither in hiPSC-derived organoids nor in the adult retina. 

### 2.3. Comparison of Different Mammal Species and in Vitro Testbeds

Photoreceptors are among the most specialized cell types in mammals. Their function and cellular maintenance are regulated by distinct transcriptional programs [26]. In addition, rods of nocturnal mammals present a unique chromatin structure, allowing vision in dim light [35]. Such ‘inverted’ chromatin structure in murine rods has been described as conferring specific regulatory activity and delaying DSB repair by NHEJ [36,37]. Here we used the transcriptome profiles of the DSB repair pathways to evaluate differences between primates, diurnal rodents, and nocturnal rodents, and evaluate a cellular in vitro model for studying DSB repair dynamics. Specifically, we investigated whether the nocturnal rod-dominated mouse retina is well-suited for testing genomic engineering approaches that are meant to treat humans. We gathered bulk RNA-seq data from adult retinas of: *Homo sapiens*, *Macaca fascicularis*, *Mus musculus*, and *Ictidomys tridecemlineatus* [38]. The curated gene lists (see Methods 4.1) were used to assess DSB repair pathway activity. Consistently, a lower expression of HDR was found in all species (Figure 4a). Moreover, human, macaque, and mouse adult retinas showed a predominant expression of NHEJ genes (Figure 4a. Wilcoxon signed-rank test: human *p*-value = 0.02; macaque *p*-value = 0.0003; mouse *p*-value = 0.01). There was a relatively low expression of the resection factors (CtIP and MRN complex) of the MMEJ pathway, with the macaque being the only exception (Figure A4). This explained the two peaks in the MMEJ distribution of the macaque, due to a high expression of *RBBP8*, which codes for the resection effector CtIP. On the other hand, the high *PARP1* expression was conserved in all species. In mice, although rods have the ‘inverse’ nuclear architecture, the DSB repair pathway transcriptomic profile correlated with the primate ones. In contrast, retinal NHEJ and MMEJ levels of ground squirrels were similar, meaning that this model system differs in DSB repair activity from humans or mice.

Based on our transcriptomic analysis, DSB repair pathways are similar in mouse and human photoreceptors. Nevertheless, testing genome editing strategies for human retinas might require different guide RNAs and donor DNA sequences. Therefore, it is important to find a well-suited postmitotic cellular model system. Photoreceptors within human retinal organoids can be used, although gene delivery and the long culturing periods are not ideal. Hence, hiPSC-derived neurons might be better suited as these protocols mimic neurogenesis in controlled and reproducible conditions in vitro [28]. As shown in Results 2.1, iNGN exited the cell-cycle after four days. We cross-correlated data from adult human retinas with iNGN during development on a DSB gene expression basis. Principal component analysis revealed that the developmental stage was the primary source of variation, grouping adult retinae together with iNGN at days 7 and 14 (Figure 4b). Spearman correlation confirmed this relationship between the samples (Figure 4c). Therefore, stem-cell-derived neurons cultured for two weeks represented an adequate and rapid model system of DSB repair for gene editing of postmitotic neurons. 

## 3. Discussion

This study represents a comprehensive assessment of DSB repair pathways by employing transcriptomic data to fully characterize DSB activity throughout retinal cell-type maturation, across cell types and species. E15 mouse scRNA-seq, together with bulk time course datasets, offered an unbiased quantification of the transcriptome of cycling and postmitotic cells. Since the three DSB pathways are competing in the cell nucleus, the amount of protection and resection factors influences the pathway choice. Here, we highlighted how gene expression levels of DSB-related genes fluctuated in the transition from dividing to non-dividing cells. Corroborating published data [39], the HDR pathway is strictly correlated to cell-cycle activity, whereas MMEJ is active at the transcriptomic level, also in postmitotic cells and NHEJ at all cell-cycle stages. Our curated gene list is helpful for assessing DSB repair pathway activity, and we used it to assess differences between rod and cone photoreceptors. Canonical retinal cell types were identified in retinal organoids by scRNA-seq analysis of four batches. In accordance with the literature, cones started to appear at day 106, whereas rods were the most abundant cell type at day 201 of culture. Rods and cones showed no differences in their transcriptomic profile of DSB repair pathways. This finding was also confirmed by analyzing published scRNA-seq data from both human and mouse adult retinas. Therefore, photoreceptors in retinal organoids represent an adequate in vitro model for testing gene editing approaches. However, their generation requires extended culturing periods of more than hundreds of days. The DSB repair pathway in the stem cell derived neuronal cellular model system correlates well with photoreceptors after 14 days in culture. Because delivery of guide RNA, donor DNA, and Cas9 into cell lines, and the subsequent analysis, is relatively easy to perform, these postmitotic cellular systems are well-suited for testing therapeutic genome editing strategies.

Comparing the whole adult retinal DSB pathway profile between different species revealed similarities between human, macaque, and mouse. Bulk transcriptomes are affected by the ensemble average. As expected from previous analysis, the NHEJ pathway has the highest expression levels. However, it is not predominant in the ground squirrel retina, which is the only cone-dominated retina in our analysis. Due to this difference in DSB repair activity, ground squirrels are likely not adequate animal models for exploring genomic engineering approaches for treating photoreceptors.

The high expression of *PARP1* for MMEJ in adult photoreceptors makes this pathway a good candidate for genome editing strategies aiming to correct IRD caused by mutations in the retina. MMEJ-based strategies enable a part of the genomic DNA to be replaced by non-random template integration. This would be strongly beneficial for IRDs (such as X-linked retinitis pigmentosa) which are characterized by one mutation hotspot [40]: The same treatment could be applied to a large number of patients [1]. Such genome editing approaches may be supported by manipulating DSB repair pathways [11], for example to enhance resection factors. However, a thorough determination of the DSB repair pathways and their regulators must be accomplished [41,42], also at the protein level. From our previous and related published work, we know that a number of DSB repair proteins, such as *53bp1*, *yH2AX*, *Ku80*, *LigIV*, are differentially expressed in mouse retinas, even at different time points after birth [37,43,44]. For this purpose, postmitotic stem cell derived neurons may also serve as an efficient cellular model system for manipulating DSB repair towards precise genome engineering.

## 4. Materials and Methods 

### 4.1. Curated Gene List for DSB Pathways

DSB repair pathways consist of different sets of proteins and complexes that can cooperate in a stochastic manner. Some proteins will only play a role in certain conditions, and others are always required [14]. Furthermore, the whole set of accessory proteins taking part in these pathways is constantly being updated [45,46]. In this study, we considered proteins which have already been described as being key in the choice of repair pathway after a DSB (Table A1). A DSB is initially recognized by two proteins called ataxia telangiectasia mutated (*ATM*) and Rad3-related protein (*ATR*) which phosphorylate *H2AX*, a member of the H2A histone family. This triggers a cascade of interactions leading to recruitment at the broken ends of either protection factors initiating NHEJ, such as p53 binding protein 1 (*TP53BP1*) and Werner syndrome RecQ-like helicase (*WRN*); or initial resection factors causing HR repair. NHEJ is initiated by the binding of the Ku70-Ku80 complex (*XRCC5-XRCC6*) to the broken ends, keeping them shielded by resection and recruiting a range of polymerases and ligases to fill the gap. Alternatively, phosphorylation of carboxy-terminal binding protein interacting protein (CtIP, coded by *RBBP8*) leads to initial resection by activating the MRN complex (*NBN-MRE11-RAD50*), generating stretches of single-strand DNA. Then, poly(ADP-ribose) polymerase 1 (*PARP1*) can bind and promote MMEJ by recruiting the DNA polymerase *θ* (*POLQ*). Alternatively, exonuclease 1 (*EXO1*) and Bloom syndrome RecQ-like helicase (*BLM*) provide additional resection, leading to single-strand annealing (SSA) if replication protein A (RPA) binds, or to HDR, in the presence of sister chromatids. In Table 1 we consider *RPA* genes together with additional resection genes, as they cooperate inhibiting MMEJ.

### 4.2. Datasets

Table 2 and Table 3 shows bulk RNA sequencing datasets and single cell RNA sequencing datasets.

### 4.3. Computational Analysis of RNA-Seq Data 

Raw reads of each available dataset were retrieved from the Gene Expression Omnibus repository.

#### 4.3.1. Bulk RNA-Seq Processing Pipeline

Primary RNA-seq data was downloaded by the fastq-dump v2.9.2 tool using each dataset’s GEO ID (Table 2) (https://ncbi.github.io/sra-tools/fastq-dump.html). Quality control, pre-processing, alignment, pseudo-alignment, and transcript-level quantification were accomplished by a self-implemented pipeline (Figure A5). Read quality was initially assessed and adapters collected using FastQC v0.11.6 (https://github.com/s-andrews/FastQC). Unfiltered reads were mapped to the respective reference sequence (Ensemble GRC38v94) for visualization and sequencing evaluation using STAR v2.5.4a [47]. The Picard v2.9.0 tool CollectRnaSeqMetrics was used to determine ribosomal, intronic, and intergenic RNA abundance (http://broadinstitute.github.io/picard). Then reads were quality filtered and trimmed using Trimmomatic v0.33, applying the recommended parameters [48]. Then, filtered reads were used for transcript-level quantification using Kallisto v0.44.0 [49]. For each species, reference indexes were built on the total collection of annotated transcript sequences (coding and non-coding) in Ensemble GrC38v94. All secondary analyses were performed on Python3 notebooks.

#### 4.3.2. Single-Cell RNA-Seq Processing Pipeline

Whenever possible, GEO ID was used to download fastq files for each dataset (Table 3). For the human adult retina (E-MTAB-74316), raw reads are not publicly available so matrix, gene, and barcode files were downloaded directly. Reads were processed following the “kallisto|bustools” workflow (Figure A5) (https://www.kallistobus.tools/about). Kallisto was initially used to pseudoalign reads to the reference transcriptome index. Then, Bustools v0.39.3 served to correct, sort, and count unique molecular identifiers (UMIs) of pseudoaligned reads, generating the count matrix of cells by genes [50]. Count matrices were loaded on Python3 notebooks and analyzed using Scanpy v1.4.5 [51]. 

#### 4.3.3. Mouse Embryo scRNA-Seq Analysis

A kallisto index was built on the set of both introns and cDNA fasta sequences. Count matrices consisting of spliced and unspliced layers were generated using the ‘kb’ command with --lamanno argument (kb-python v0.24.4 package) and using the 10×.v2 barcode list. Two matrices were generated (one per sequencing batch) and concatenated. Cells were filtered to have more than 2000 UMI counts, and between 400 and 3500 genes (to exclude doublets), resulting in a matrix composed of 4184 cells per 22,730 genes. Counts were normalized and log-transformed. The top 1000 highly-variable genes were considered for batch correction by matching mutual nearest neighbor and dimensionality reduction using principal components analysis [52]. The neighbor graph was computed on the 50 principal components by batch-balanced k nearest neighbor for a homogeneous integration between batches [53]. A neighbors graph was then used to impute clusters using a louvain algorithm [54], and visualized using UMAP [55]. The clusters were annotated following previously-described markers [22]. With our analysis we were able to reproduce the findings previously described. Velocity vectors were estimated using the scvelo v0.1.24 package implementation (https://scvelo.readthedocs.io/index.html) [56], employing the dynamical model to fully determine the dynamics of splicing kinetics in all genes. In brief, after isolating the photoreceptor lineage from the whole dataset we computed moments, velocities, and a velocity graph, which we visualized embedded on a UMAP.

Scoring the DSB repair pathways was done by averaging the raw gene counts of each pathway. Due to many dropouts in scRNA-seq, it was not possible to capture these genes in some cells, resulting in a score of 0. Those cells were ignored in the comparison analyses.

#### 4.3.4. Cell-Cycle and DSB Pathway Correlation in Bulk Time Course RNA-Seq

Count matrices reporting each time point and replicate gene expression were quality checked using a correlation matrix plot showing dynamics of development. Genes were annotated according to the DSB curated lists and log-transformed TPM (transcripts per million) were used for plotting and analysis.

#### 4.3.5. Comparison of Mammal Retina and iNGN

Count matrices of human, macaque, ground squirrel, and mouse gene expression were loaded to a Python notebook. Counts were quantile-normalized across replicates within the same species, and log transformed. Genes of the same pathways were then annotated using our curated lists. These matrices were then used both for plotting and statistical test.

To compare human and iNGN repair pathways, merged counts of human and iNGN replicates were preprocessed, and the subsets of DSB genes were used as features for principal component analysis and Spearman correlation.

#### 4.3.6. Organoid scRNA-Seq Analysis 

Count matrices for each sequencing batch (d64, d106, d201, d330) were obtained as described in Methods 4.3.2. The four matrices, consisting of 45,547 sequenced cells, were initially concatenated. Only cells expressing between 500 and 7000 genes, fewer than 20,000 UMIs, and fewer than 0.1% of mitochondrial counts were kept in the filtering. The matrix at this point comprised 26,707 cells and 26,738 genes, with a median number of genes per cell of 2019. Counts were normalized and log transformed, and the 600 highly variable genes in common between the four batches were considered for batch correction (mutual nearest neighbor) and dimensionality reduction using PCA. The first 40 principal components were used to compute the neighbor graph using batch-balanced k nearest neighbor, and subsequently for clustering using the Leiden algorithm and UMAP visualization. Unbiased clusters were computed via t-test of each group against all the others. Those markers were used to calculate an overlap score with known retinal cell-type markers. In this way, we could annotate clusters according to the calculated unbiased markers. Pseudotime was computed on a subset of the whole dataset, keeping progenitors and neuron-fate clusters. DSB scores were computed by averaging the raw counts of genes in part of each pathway. Cells with a score of 0 were ignored in the visualization and pairwise comparison. For the latter, t-test was employed after checking for normal distribution of the data by Shapiro Wilk test.

### 4.4. Generation of hiPSC-Derived Retinal Organoids 

hiPSC lines used to generate retinal organoids were produced as previously described [57], and tested for the presence of stemness markers as well as germ-layer differentiation potential. hiPSCs were cultured in tissue culture-treated plates (BD Biosciences, USA), coated with Matrigel hESC-Qualified Matrix (Corning, USA) at 5% O_2_, 5% CO_2_, and 37 °C. hiPSCs were maintained using FTDA medium [58]. Regions of differentiation were mechanically removed by scraping. All procedures were in accordance with the Helsinki convention and approved by the Ethical Committee of the Eberhard Karls University, Tübingen (Nr. 678/2017BO2). Control persons gave their written consent.

hiPSC-derived retinal organoids were generated as previously shown [59], following a described protocol [31]. Briefly, on day 0, hiPSCs were dissociated as single cells using TrypLE (Thermo Fisher Scientific, Waltham, MA, USA), resuspended in PeproGrow hESC medium (PeproTech, Hamburg, Germany), and 10,000–30,000 cells per well were distributed in non-adherent v-shaped 96-well plates (Sarstedt, Nümbrecht, Germany) where they re-aggregated generating embryoid bodies (EBs). These were grown for 7 days in neural induction medium (NIM) composed of DMEM/F-12 (1:1) GlutaMAX Supplement (Thermo Fisher Scientific, USA), supplemented with 24 nM sodium selenite (Sigma-Aldrich, St. Louis, MO, USA), 16 nM progesterone (Sigma-Aldrich, USA), 80 µg/ml human holotransferrin (Serologicals, USA), 20 µg/ml human recombinant insulin (Sigma-Aldrich, USA), 88 µM putrescine (Sigma-Aldrich, USA), 1× minimum essential media non-essential amino acids solution (NEAA; Thermo Fisher Scientific, USA), 1× antibiotic-antimycotic (AA; Thermo Fisher Scientific, USA). On day 7, embryoid bodies were seeded onto tissue culture-treated 6-well plates (BD Bioscience, San Jose, CA, USA) coated with matrigel growth factor reduced basement membrane matrix (Corning, New York, NY, USA) at a density of 32 EBs per well. The medium was replaced every other day. On day 16, the medium was switched from NIM to B27-based differentiation medium (BRDM) composed of DMEM/F-12 (3:1) GlutaMAX supplement, supplemented with 1× B-27 supplement (without vitamin A; Thermo Fisher Scientific, USA), 1× NEAA, and 1× AA. On day 24, the retinal field areas were manually detached and collected in 10 cm bacterial petri dishes (Greiner Bio-One, Rainbach, Austria). Starting from day 40, the BRDM medium was supplemented with 10% fetal bovine serum (FBS, Thermo Fisher Scientific, USA) and 100 µM taurine (Sigma Aldrich, USA). Between day 70 and day 100, 1 µM retinoic acid (Sigma Aldrich, USA) was added and subsequently reduced to 0.5 µM between day 100 and day 190. The medium was changed twice a week until the respective experimental endpoints. Non-retinal tissue, as well as retinal pigmented epithelial cells (RPEs) were excised from the retinal organoid spheres during their first weeks of harvesting in suspension. All the differentiation steps were cultured in normoxia.

### 4.5. Preparation and Sequencing of hiPSC-Derived Retinal Organoid Single-Cell cDNA Libraries 

For each time point, 4–5 retinal organoids were selected and dissociated using the Neurosphere Dissociation Kit (P) (Miltenyi Biotec, Bergisch Gladbach, Germany) according to the manufacturer’s instructions. Cell suspensions were filtered using a 30 µm cell strainer (Miltenyi Biotec, Germany). Cell numbers were assessed using a Neubauer counting chamber and viability was determined using 0.2% trypan blue (Sigma-Aldrich, USA). Single-cell gene expression libraries were generated using the 10× Chromium Next GEM Single Cell 3’ Reagent Kit v3.1 (10× Genomics, USA) according to the manufacturer’s instructions. In brief, approximately 18,000 (d64), 16,000 (d106), 18,000 (d201), and 10,000 cells (d330) were loaded on the Chromium Next GEM Chip G (10× Genomics, USA) and run on the Chromium Controller (10× Genomics, USA) to be partitioned into gel bead in emulsion (GEMs). Cell lysis and reverse transcription occurred within the GEMs, resulting in cDNA from poly-adenylated mRNA containing GEM-specific barcodes as well as transcript-specific unique molecular identifiers (UMIs). After breaking the emulsion, cDNA was amplified in 11 cycles, enzymatically fragmented, end-repaired, extended with 3’ A-overhangs, ligated to adapters, and finally amplified via PCR in 13 cycles while adding the P5 and P7 sequences needed for Illumina bridge amplification, as well as a sample indices (Chromium i7 Multiplex kit; 10× Genomics, USA). The concentration of the final scRNA libraries was determined using the Qubit dsDNA HS Assay Kit (Thermo Fisher Scientific, USA), and their fragment size was determined using the Bioanalyzer High Sensitivity DNA Kit (Agilent Technologies, USA). All scRNA libraries were pooled and paired-end sequenced on the Illumina NovaSeq 6000 platform (Illumina, San Diego, CA, USA) using 28 cycles for read 1, 91 cycles for read 2, and 8 cycles for the i7 index. 

### 4.6. Retinal Organoid Immunocytochemistry

Retinal organoids used for immunocytochemistry were fixed in 4% paraformaldehyde (PFA; AppliChem GmbH, Darmstadt, Germany) for 20 minutes at room temperature, washed once with PBS, and immersed in 30% sucrose (Sigma-Aldrich, USA) overnight. Subsequently, samples were embedded using Tissue-Tek O.C.T. (Sakura Finetek, Alphen aan den Rijn, Netherlands) inside cryo molds, frozen on a precooled metal plate and stored at −80 °C. Next, the samples were sectioned (14 µm) using a cryostat.

For immunocytochemistry, cryosections were rehydrated with PBS for 10 minutes, blocked, and permeabilized for 1 hour at room temperature using 10% normal donkey serum in PBS supplemented with 0.2% Triton X-100 (Carl Roth, Karlsruhe, Germany). Primary antibodies were diluted in the blocking solution and applied to the samples at 4 °C overnight. The sections were then washed three times with PBS before the secondary antibody solution was added for 1 hour at room temperature. The secondary antibodies were diluted in 5% normal donkey serum in PBS supplemented with 0.1% Triton X-100. Finally, the sections were washed five times with PBS and mounted using ProLong Gold Antifade Reagent with DAPI (Thermo Fisher Scientific, USA).

Image stacks were acquired using an LSM 710 confocal microscope (Carl Zeiss, Jena, Germany).

Primary antibodies list:

Arrestin 3 (ARR3) (goat, 1:50, sc54355, Santa Cruz Biotechnologies, Dallas, TX, USA)

Guanine nucleotide-binding protein G(t) subunit alpha-1 (GNAT1) (rabbit, 1:500, sc389, Santa Cruz Biotechnologies, Dallas, TX, USA)

Secondary antibodies list:

Donkey anti-rabbit IgG (H + L) Alexa Fluor® 488/568 (1:250, Thermo Fisher Scientific, USA)

Donkey anti-goat IgG (H + L) Alexa Fluor® 647 (1:250, Thermo Fisher Scientific, USA)

## Figures and Tables

**Figure 1 ijms-21-01380-f001:**
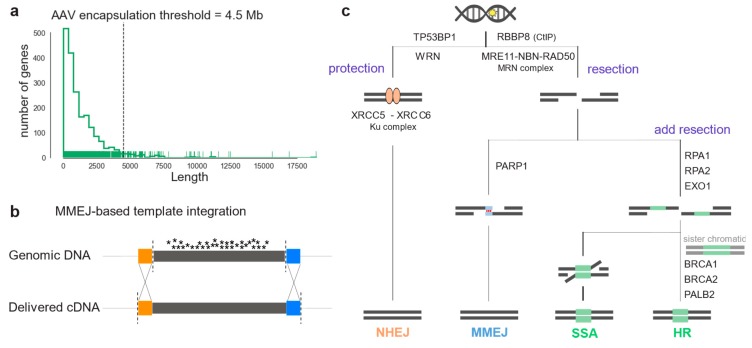
Large genomic portions can be corrected by in vivo genome editing relying on endogenous DSB repair pathways activity. (**a**) Coding sequence length of IRD genes listed in RetNet. The dashed line indicates the maximum cargo limit for AAV transfer. (**b**) Scheme illustrating homology-dependent genome engineering (HDR or MMEJ) to replace an entire mutation hotspot exon. Colored boxes (orange and blue) indicate homology regions. (**c**) DSB repair pathway diagram illustrating the enzymes listed in Table 1. After a DSB in the genomic DNA, the broken ends can be resectioned or protected. Protection by *TP53BP1* or *WRN*, and Ku complex leads to NHEJ repair (orange). On the other hand, resection forms single-strand DNA overhangs that can reveal homology. Binding of *PARP1* and the presence of micro-homology leads to MMEJ repair (blue). If resection is prolonged, repair is by SSA or HDR depending on the cell-cycle stage (green).

**Figure 2 ijms-21-01380-f002:**
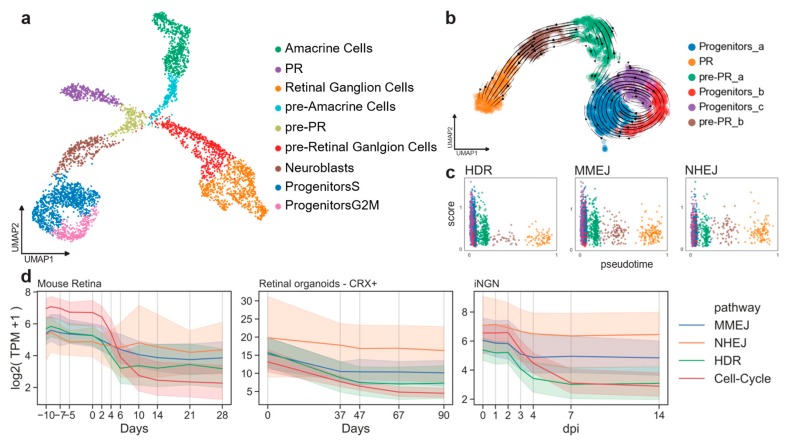
Cell-cycle-related gene expression correlates to DSB pathways. (**a**) E15 mouse retinal cells in two UMAP dimensions showing clusters of known amacrine cells, retinal ganglion cells and PR cells at this developmental stage. (**b)** Subset of the developmental PR trajectory in two UMAP dimensions. Arrows represent the vector calculated for each cell according to the RNA velocity approach. (**c**) Pseudo-timed HDR, MMEJ, and NHEJ scores within the photoreceptor developmental trajectory. Cell colors refer to clusters in b). (**d**) DSB repair pathway activity during cell-cycle exit. Lines represent the mean value, and the error bands the 95% confidence intervals.

**Figure 3 ijms-21-01380-f003:**
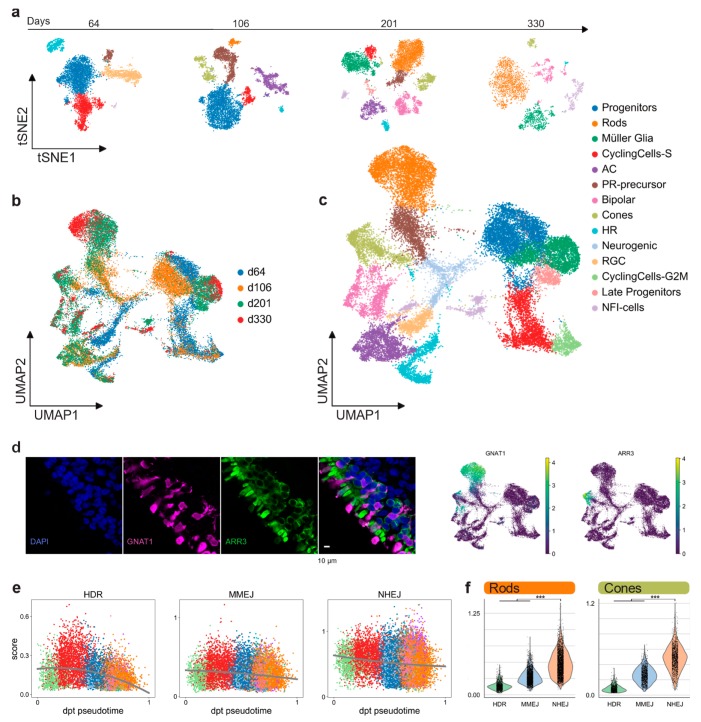
hiPSC-derived retinal organoids. (**a**) hiPSC-derived organoid sequenced cells visualized in two t-sne dimensions, showing different cell-type composition at four different time points. (**b**) 26,700 quality filtered cells merged together and colored by batch, visualized in two-dimensional UMAPs. (**c**) Merged retinal organoid dataset visualized in two-dimensional UMAPs and colored by annotated retinal clusters. (**d**) Immunohistochemistry indicates, after scRNA-seq analysis, the presence of distinct rod (*GNAT1*) and cone (*ARR3*) populations in the outer layer of the hiPSC-derived retinal organoids. (**e**) Pseudo-timed HDR, MMEJ, and NHEJ scores within the neuronal lineage of hiPSC-derived organoids. Lines represent the polynomial fit to the data. (**f**) DSB repair pathway activity scores of rods and cones in hiPSC-derived retinal organoids.

**Figure 4 ijms-21-01380-f004:**
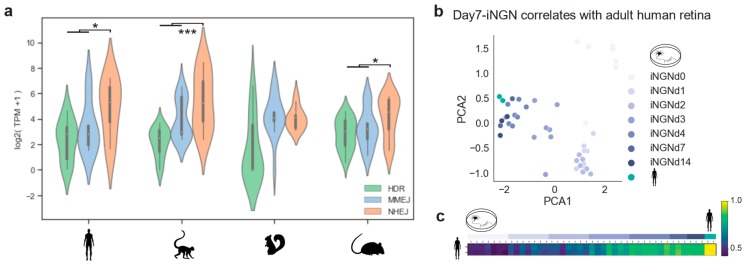
Cell-cycle-related genes correlate to DSB pathways. (**a**) Violin plot of gene expression abundance in DSB repair pathways in primates and rodents. Pairwise comparison significance was tested by Wilcoxon signed-rank test. (**b**,**c**) PCA plot and Spearman correlation of all sequencing replicates showing the relationship between DSB repair pathway genes in human retinas and developing in vitro neurons.

**Table 1 ijms-21-01380-t001:** Curated double-strand break repair pathways genes. This table lists key components of DSB repair pathways as discussed in Methods 4.1.

Gene Symbol (Mouse Name)	Gene Name	Pathway	Function
*XRCC5*	*X-ray repair cross complementing 5* *(Ku80)*	*NHEJ*	*determinant*
*XRCC6*	*X-ray repair cross complementing 6* *(Ku70)*	*NHEJ*	*determinant*
*TP53BP1 (Trp53bp1)*	*Tumor protein p53 binding protein 1*	*NHEJ*	*protection*
*WRN*	*Werner syndrome RecQ-like helicase*	*NHEJ*	*protection*
*PARP1*	*Poly(ADP-ribose) polymerase 1*	*MMEJ*	*determinant*
*RBBP8*	*RB binding protein 8, endonuclease* *(CtIP)*	*MMEJ*	*resection*
*MRE11 (Mre11a)*	*Meiotic recombination 11 homolog A*	*MMEJ*	*resection*
*NBN*	*Nijmegen breakage syndrome 1*	*MMEJ*	*resection*
*RAD50*	*Double-strand break repair protein*	*MMEJ*	*resection*
*BRCA1*	*Breast cancer type 1 susceptibility protein*	*HDR*	*determinant*
*BRCA2*	*Breast cancer type 2 susceptibility protein*	*HDR*	*determinant*
*RAD51*	*BRCA1/BRCA2-containing complex, subunit 5*	*HDR*	*determinant*
*PALB2*	*Partner and localizer of BRCA2*	*HDR*	*determinant*
*RPA1*	*Replication protein A1*	*HDR*	*SSA*
*RPA2*	*Replication protein A2*	*HDR*	*SSA*
*EXO1*	*Exonuclease 1*	*HDR*	*Add-resection*
*BLM*	*Bloom syndrome RecQ-like helicase*	*HDR*	*Add-resection*

**Table 2 ijms-21-01380-t002:** Bulk RNA sequencing datasets.

GEO ID	Species	Sample		Time Points (Days)
*GSE84930*	*Homo sapiens*	*in vivo*	*Retina*	*Adult*
*GSE84927*	*Mus musculus*	*in vivo*	*Retina*	*Adult*
*GSE84929*	*Macaca fascicularis*	*in vivo*	*Retina*	*Adult*
*GSE84931*	*Ictidomys tridecemlineatus*	*in vivo*	*Retina*	*Adult*
*GSE101986*	*Homo sapiens*	*in vitro*	*Organoids CRX+ cells*	*d0, d37, d47, d67, d90*
*GSE118307*	*Homo sapiens*	*in vitro*	*Neurons*	*d0-4, d7, d14*
*GSE101986*	*Mus musculus*	*in vivo*	*Retina*	*E11-P28*

**Table 3 ijms-21-01380-t003:** Single cell RNA sequencing datasets.

GEO ID	Species	Sample		Sequencing	Age
*GSE130636*	*Homo sapiens*	*in vivo*	*Retina*	*10x v3*	*Adult*
*E-MTAB-74316*	*Homo sapiens*	*in vivo*	*Retina*	*10x v2*	*Adult*
*GSE63473*	*Mus musculus*	*in vivo*	*Retina*	*DropSeq*	*Young*
*GSE122566*	*Mus musculus*	*in vivo*	*Retina*	*10x v2*	*Embryo (E15)*
*-*	*Homo sapiens*	*in vitro*	*Organoid*	*10x v3*	*d64, d106, d220, d330*

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
