# Peer review of "Using Transcriptomic Analysis to Assess Double-Strand Break Repair Activity: Towards Precise in Vivo Genome Editing"

_ijms, 2020, doi:10.3390/ijms21041380_

Round 1
Reviewer 1 Report
The authors of the manuscript entitled “Using transcriptomic analysis to assess double-strand break repair activity: towards precise in vivo genome editing”, performed transcriptomic analysis of the retinas of several species either as a bulk or single cell. With this information, authors studied whether the genes encoding for proteins involved in genome editing repair pathways are expressed or not. In addition, the authors show that organoids can be suitable models to develop genomic engineering approaches.
The manuscript is well written and figures are well prepared. There are few small minor comments to be addressed:
1) The authors should ensure that all gene names are in italics (sometimes they are not).
2) in vitro and in vivo should be written without the hyphen.
3) I would suggest to call the Appendix A -> Supplementary material. And figures and tables S1 S2 etc. This will make the manuscript read easier.
4) Figure A4 and A5 miss figure legend. It would be good to add some lines that can help to interpret the results.
5) There are a few typo’s:
line 289: should read “updated”
Table A1: RHO is listed twice. Perhaps it would be good to order the markers in alphabetical order
Table A1: Title should read: Gene markers used for overlapping annotation in organoids.
Figure A3: I do not think the in vivo is needed (if it is adult retina is Obvious that comes from an animal/individual), although then I would say it should be ex vivo.
Author Response
We would like to thank Reviewer 1 for the kind words and appreciation of our work. Please find below our point-by-point responses interspersed in the reviewer comments.
1) The authors should ensure that all gene names are in italics (sometimes they are not).
We checked the entire manuscript and used italics for all gene names. All changes can be found in the revised document in Word’s track mode.
2) in vitro and in vivo should be written without the hyphen.
We agree and have done all changes.
3) I would suggest to call the Appendix A -> Supplementary material. And figures and tables S1 S2 etc. This will make the manuscript read easier.
We have strictly followed the author guidelines. In the case the editors agree, we are happy to change “Appendix” to “Supplementary material”. We agree with Reviewer 1 that this nomenclature would be more intuitive.
4) Figure A4 and A5 miss figure legend. It would be good to add some lines that can help to interpret the results.
The legends have been accidentally removed in our last version that we have submitted. We have added the legends to the revised manuscript.
5) There are a few typo’s:
We would like to thank Reviewer 1 for the thorough review. We have corrected all typos.
line 289: should read “updated”
Table A1: RHO is listed twice. Perhaps it would be good to order the markers in alphabetical order
Table A1: Title should read: Gene markers used for overlapping annotation in organoids.
Figure A3: I do not think the in vivo is needed (if it is adult retina is Obvious that comes from an animal/individual), although then I would say it should be ex vivo.
Reviewer 2 Report
Title: “Using transcriptomic analysis to assess double-strand break repair activity: towards precise in-vivo genome editing”
Authors provide an in-depth analysis of the transcriptional signature of retinal cell populations in relationship with double-strand break repair pathways. The strength of this study is the combination of multiple models, from in vitro 2D and 3D cultures to in vivo samples, including different species. Their analysis of transcriptomics and single cell RNA seq data to characterize DSB pathway offer a support for previous literature and reinforces the possibility of developing therapeutic in vivo genome editing in post-mitotic cells.
The highlight of this manuscript is the characterization of 2D cultured induced neurons, which show similar transcriptional profiles than in vivo samples. This suggests that 2D cultured neurons could be a good model to test gene therapy approaches.
Comments:
-Authors investigate transcriptional levels of DSB-related genes. Showing differential expression at the protein level would provide added value and further proof of activation of those pathways (maybe in in vitro models upon induced DNA damage?).
-Line 190-194, Figure 3d. Any comment on the different (or not) scores between MMEJ and NHEJ?
-Line 255. Authors state “MME is active”. Is transcriptomics enough to claim that a pathway is active?
-Any clue about the preferential repair pathway in photoreceptors? Both MMEJ and NHEJ seem to be expressed at high levels. Is it equal? MMEJ-related genes are indeed expressed, but is it more efficient than NHEJ in repairing DSB?
-More details on the statistics: Why t-test? Parametric or non-parametric? Does data follow normal distribution?
-Any comment or explanation on the unexpected or expected profile of squirrels vs. mice? From the text, the reader could expect that squirrels would have a similar profile as other diurnal mammals. [Fig 4a and lines 269-273]
Minor comments:
-Does IJMS have guidelines regarding the spelling of in vitro and in vivo? I have never seen that written with a hyphen [in-vitro and in-vivo].
-Lines 88-105 state the objectives and results. I believe this section needs some re-writing or clarity to exactly highlight the new results obtained during this study and not to just mention the different in vitro/in vivo models and which type of data were used.
-Line 197: “NHEJ factors are expressed most, and MMEJ less.” What does that sentence mean? Grammar mistake?
-Line 403: “seeded onto a well plate”, please indicate what well plate (for example a 9.6 cm2 well plate)
Author Response
We would like to thank Reviewer 2 for the assessment of our work. Please find below our point-by-point responses interspersed in the reviewer comments. All changes can be found in the revised document in Word’s track mode.
Comments:
-Authors investigate transcriptional levels of DSB-related genes. Showing differential expression at the protein level would provide added value and further proof of activation of those pathways (maybe in in vitro models upon induced DNA damage?).
We totally agree with Reviewer 2 that protein levels need to be studied as well. Within this article, we specifically focused on transcriptomic data because of available RNA-Seq data for a comparative study. However, from our previous and related published work, we know that a number of DSB repair proteins (e.g. 53bp1, yH2AX, Ku80, Lig IV) are differentially expressed in mouse retinas even at different time points after birth (Frohns et al, Curr Biol 2014 PMID: 24794298, Müller et al, Front Neurosci 2018 PMID: 29765300, Sahaboglu et al 2017 PMID: 28723922). In addition, we have planned follow-up studies for DSB repair pathway manipulation in in vitro models for which protein data as readout are required. We have added this information to the discussion.
-Line 190-194, Figure 3d. Any comment on the different (or not) scores between MMEJ and NHEJ?
Figure 3d refers to the presence of both cones and rods in retinal organoids. Probably Reviewer 2 is commenting on figure 3e or 3f. NHEJ is described in the literature to be the predominantly active pathway in mammal cells. Therefore, it is not a surprise to find NHEJ genes highly expressed in photoreceptors. We are now highlighting the persistent high levels of MMEJ and added “up to post-mitotic stage” to the text (line 212).
-Line 255. Authors state “MME is active”. Is transcriptomics enough to claim that a pathway is active?
We agree with Reviewer 2 that our statement was not sufficiently supported by just transcriptomic data. We have revised and clarified this statement to “...MMEJ is active at the transcriptomic level …”
-Any clue about the preferential repair pathway in photoreceptors? Both MMEJ and NHEJ seem to be expressed at high levels. Is it equal? MMEJ-related genes are indeed expressed, but is it more efficient than NHEJ in repairing DSB?
Currently, it is unknown whether there exists a preferential DSB repair pathway in photoreceptors. Therefore, it is difficult to judge which pathway is more active in mature photoreceptors. It is known that NHEJ is active and at work in photoreceptors (Yu et al 2017, PMID: 28291770). We are currently comparing NHEJ and MMEJ activity in photoreceptors in vivo and in the 2D cell culture in order to address this question.
-More details on the statistics: Why t-test? Parametric or non-parametric? Does data follow normal distribution?
We thank Reviewer 2 for pointing this out. In the whole study we used Student t-test, which by definition is a parametric statistical test. Parametric tests assume that the data follow a normal distribution. We checked for normal distribution by the Shapiro Wilk test. As expected, all pathway scores within cones and rods from scRNA dataset follow normal distribution (Figure 3e and Figure A3). Therefore in these comparisons, Student t-test is reliable to assess . On the other hand, in the comparison of DSB scores in mammals (Figure 4), not all the distributions are normal. According to the Shapiro Wilk test, MMEJ and HDR of the ground squirrel, and NHEJ of human, do not follow normal distribution. In this case a non-parametric test to compare two samples would suit better, i.e. the Wilcoxon signed-rank test. We applied such statistical tests on all the comparisons of this section, and results remain the same. Yet we modified the main text, indicating the non-parametric test used and the new p-values.
-Any comment or explanation on the unexpected or expected profile of squirrels vs. mice? From the text, the reader could expect that squirrels would have a similar profile as other diurnal mammals. [Fig 4a and lines 269-273]
Squirrels have 80% cone distribution in their retina, while mice only have 3-5% cones among 95% rods. We observed a similar expression level of NHEJ and MMEJ in squirrel retina. The MMEJ pathway consists of resection factors and PARP1. While PARP1 is expressed as high as NHEJ in all compared retinal samples, the resection factors from squirrel data show higher expression levels (Figure A4). However, we cannot currently explain this finding in detail, it is likely species specific. Through our single-cell analysis, we haven’t detected any relevant differences of DSB repair pathways between rods and cones in both human and mouse. We have updated the results and discussion sections.
Minor comments:
-Does IJMS have guidelines regarding the spelling of in vitro and in vivo? I have never seen that written with a hyphen [in-vitro and in-vivo].
We have eliminated all hyphens.
-Lines 88-105 state the objectives and results. I believe this section needs some re-writing or clarity to exactly highlight the new results obtained during this study and not to just mention the different in vitro/in vivo models and which type of data were used.
We would like to thank the Reviewer, we have updated this paragraph.
-Line 197: “NHEJ factors are expressed most, and MMEJ less.” What does that sentence mean? Grammar mistake?
We apologize for this confusion. The sentence was an artefact from an edited version. We have now deleted this sentence.
-Line 403: “seeded onto a well plate”, please indicate what well plate (for example a 9.6 cm2 well plate)
We have added following information to the methods: “Tissue culture- treated 6-well plates (BD Bioscience, USA)”